# Preconditioning or Postconditioning with 8-Br-cAMP-AM Protects the Heart against Regional Ischemia and Reperfusion: A Role for Mitochondrial Permeability Transition

**DOI:** 10.3390/cells10051223

**Published:** 2021-05-17

**Authors:** Igor Khaliulin, Raimondo Ascione, Leonid N. Maslov, Haitham Amal, M. Saadeh Suleiman

**Affiliations:** 1Institute for Drug Research, School of Pharmacy, Faculty of Medicine, Hebrew University of Jerusalem, Pharmacy Building, Ein Karem, Jerusalem 91120, Israel; Haitham.amal@mail.huji.ac.il; 2Bristol Medical School (THS), Faculty of Health Sciences, University of Bristol, Bristol Royal Infirmary, Upper Maudlin Street, Bristol BS2 8HW, UK; R.Ascione@bristol.ac.uk (R.A.); M.S.Suleiman@bristol.ac.uk (M.S.S.); 3Cardiology Research Institute, Tomsk National Research Medical Center, The Russian Academy of Sciences, 111 a, Kievskaya Street, 634012 Tomsk, Russia; maslov@cardio-tomsk.ru

**Keywords:** cyclic AMP, heart, regional ischaemia, cardioprotection, reperfusion injury, mitochondria permeability transition pore, hexokinase II

## Abstract

The cAMP analogue 8-Br-cAMP-AM (8-Br) confers marked protection against global ischaemia/reperfusion of isolated perfused heart. We tested the hypothesis that 8-Br is also protective under clinically relevant conditions (regional ischaemia) when applied either before ischemia or at the beginning of reperfusion, and this effect is associated with the mitochondrial permeability transition pore (MPTP). 8-Br (10 μM) was administered to Langendorff-perfused rat hearts for 5 min either before or at the end of 30 min regional ischaemia. Ca^2+^-induced mitochondria swelling (a measure of MPTP opening) and binding of hexokinase II (HKII) to mitochondria were assessed following the drug treatment at preischaemia. Haemodynamic function and ventricular arrhythmias were monitored during ischaemia and 2 h reperfusion. Infarct size was evaluated at the end of reperfusion. 8-Br administered before ischaemia attenuated ventricular arrhythmias, improved haemodynamic function, and reduced infarct size during ischaemia/reperfusion. Application of 8-Br at the end of ischaemia protected the heart during reperfusion. 8-Br promoted binding of HKII to the mitochondria and reduced Ca^2+^-induced mitochondria swelling. Thus, 8-Br protects the heart when administered before regional ischaemia or at the beginning of reperfusion. This effect is associated with inhibition of MPTP via binding of HKII to mitochondria, which may underlie the protective mechanism.

## 1. Introduction

Acute myocardial infarction (AMI) remains the most significant cause of death and disability worldwide [1]. It is commonly accepted that the only effective treatment of AMI is reperfusion of the ischaemic area, which can be attained by thrombolysis, percutaneous coronary intervention, or coronary artery bypass grafting [2,3,4]. Unfortunately, reperfusion injury can cause necrosis and apoptosis of the affected myocardium [5]. Reperfusion injury is associated with the opening of the mitochondria permeability transition pore (MPTP) which can be caused by elevated concentrations of Ca^2+^ [6] and reactive oxygen species (ROS) [7,8]. Currently, there are no effective interventions that can reduce the size of the infarction or attenuate remodelling towards heart failure.

We and others have shown that the cyclic AMP (cAMP) signalling pathways are cardioprotective under different conditions [9,10,11]. A major problem associated with these interventions, however, is their reliance on of β-adrenergic receptors (βARs). The sensitivity of these receptors could be blunted in chronic heart failure or hypertension due to βAR hyperstimulation mediated by the sympathoadrenal system [12,13]. Therefore, direct activation of the protective cAMP signalling that does not involve βAR stimulation would overcome this problem. cAMP activates two major signal transduction pathways in the heart, PKA [14] and a guanine nucleotide exchange protein directly activated by cAMP (Epac) [15]. We recently found that a brief (5 min) perfusion of hearts with the non-selective cell-permeable cAMP analogue 8-Br-cAMP-AM (8-Br) at the concentration of 10 μM can induce a potent cardioprotective effect [16]. However, experiments in that study were performed on hearts exposed to global ischaemia/reperfusion (I/R). Also, the cAMP analogues in that study were applied only before ischaemia. The clinical applicability of this kind of treatment is limited to cardiac surgery involving aortic cross-clamping that causes global ischaemia. Meanwhile, the aetiology of AMI is mostly associated with regional ischaemia caused by occlusion of a coronary artery [17]. Furthermore, the treatment prior to the ischemic event is impossible in most cases because an incident of AMI cannot be predicted. Therefore, a drug is needed that can protect the heart against regional I/R and when applied at the onset of reperfusion (recanalization of the coronary artery). In this work, we studied the cardioprotective potential of 8-Br when administered before regional ischaemia or at the start of reperfusion following regional ischaemia. Since MPTP opening triggered by high levels of Ca^2+^ and ROS is the critical mechanism of reperfusion injury [18], we investigated the effect of 8-Br on the sensitivity of MPTP to Ca^2+^ overload.

## 2. Materials and Methods

### 2.1. Materials

The non-selective cell-permeable cAMP analogue 8-bromoadenosine-3’,5’-cyclic monophosphate, acetoxymethyl ester (8-Br) was obtained from BioLog Life Science Institute (Bremen, Germany). Other chemicals were acquired from Sigma (Gillingham, UK). General consumables were bought from Fischer Scientific (Loughborough, UK) or VWR-Jencons (Lutterworth, UK).

### 2.2. Animals

The experiments were carried out using two-month-old male Wistar rats (250–260 g) obtained from Charles River Laboratories (Oxford, UK). Before experiments, the rats were housed at the University of Bristol Animal Services Unit for 2–4 days at the standard conditions: air temperature of 23 ± 1 °C, relative humidity of 60–70%, light/dark cycle of 12 h, and free access to standard rat chow and water. The rats were kept in standard cages with aspen bedding. Two to three animals were present in a cage. The animals were sacrificed by stunning and cervical dislocation and the hearts were extracted for Langendorff perfusion.

### 2.3. Langendorff Perfusion of Isolated Rat Heart

Rats were killed as described above. Hearts (~0.75 g) were immersed in ice-cold Krebs–Henseleit buffer (KH) containing (mM): 118 NaCl, 25 NaHCO_3_, 4.8 KCl, 1.2 KH_2_PO_4_, 1.2 MgSO_4_, 11 glucose, and 1.2 CaCl_2_, gassed with 95% O_2_–5% CO_2_ at 37 °C (pH 7.4) and perfused by Langendorff as described previously [19]. The equilibration period lasted for 30 min before the treatment. Acquisition of the data followed by the data analysis was performed using a PowerLab System (ADInstruments Ltd., Oxford, UK). The following parameters were calculated: left ventricular developed pressure (LVDP) was computed as the difference between left ventricular systolic pressure (LVSP) and left ventricular end-diastolic pressure (LVEDP); rate-pressure product (RPP) was calculated as the product of LVDP and heart rate (HR); time derivatives of pressure during contraction (+dP/dt) and relaxation (−dP/dt) were calculated by Chart 5 software (ADInstruments Ltd., Oxford, UK). Preischaemic values of the parameters of haemodynamic function were assessed at the end of the equilibration period before a preischaemic treatment. Since the perfused hearts beat spontaneously, haemodynamic function depended on both LVDP and HR. Therefore, RPP was used as the main parameter of haemodynamic function. The hearts with an RPP less than 15,000 mmHg·beat·min^−1^ at the end of the equilibration period were withdrawn from the experiment. For analysis of the dynamics of RPP and LVDP during regional ischaemia and reperfusion, the results were presented as a percentage of pre-treatment value. This normalization was carried out to avoid significant variations in the absolute values.

### 2.4. Heart Perfusion Protocols

The experimental protocols are shown in Figure 1. After a 40 min preischaemic period, hearts were subjected to 30 min regional normothermic ischaemia (37 °C) followed by 2 h reperfusion. Regional ischaemia was induced by occlusion of the left anterior descending coronary artery (LAD) [20]. Briefly, a 5/0 braided silk thread was placed around the LAD. The ligature was positioned at the starting quarter of LAD. Reperfusion was induced by the release of the thread around the LAD.

Hearts were randomly assigned to control (C) or intervention (8-Br) groups. In the first intervention group, 8-Br (10 μM) was administered for 5 min, followed by 5 min washout before regional ischaemia and reperfusion (8-Br-Pre group, *n* = 5). In the second intervention group, 5 min heart perfusion with 10 μM 8-Br and 5 min washout were carried out at the end of regional ischaemia (8-Br-Post group, *n* = 7). Hearts of control groups (C-Pre, *n* = 7, and C-Post, *n* = 6) were not subjected to any pharmacological intervention. At the end of the experimental protocol, hearts were used for quantification of the infarct size. Other hearts for control (C-Mito, *n* = 5) and interventions (8-Br-Mito, *n* = 7) were used for the determination of hexokinase II (HKII) binding to mitochondria and Ca^2+^-induced mitochondria swelling.

### 2.5. Analysis of the Left Ventricular Arrhythmias

Electrocardiogram (ECG) was recorded over the duration of the experiment by placing two 0.1 mm in diameter silver wire electrodes on the heart apex and right atrium. The ground electrode of the ECG was attached to the base of the aorta. The electrodes were connected to a bio amplifier of a PowerLab System. The ventricular arrhythmias were analysed for the duration of the LAD occlusion and the first 30 min of reperfusion according to the Lambeth Conventions [21]. The incidences of the following types of arrhythmias were assessed: ventricular premature beats (VPBs), ventricular tachycardia (VT—four or more consecutive VPBs), and ventricular fibrillation (VF—irregular deflections of QRS). Bigeminy and salvos (VPB doublet or triplet) were included in the analysis of VPBs. We used the scoring system of Curtis and Walker (1988) [22] with modifications to quantify arrhythmias. The following characteristics of the scoring system were adopted: 0, no arrhythmia; 1, 1–100 VPBs; 2, >100 VPBs; 3, <three episodes of VF/VT; 4, 3–10 episodes of VF/VT; and 5, >10 episodes of VF/VT. After 30 min reperfusion, ECG was largely restored and no malignant arrhythmias were observed at this stage of the experiment.

### 2.6. Infarct Size

The infarct size measurements were carried out at the end of reperfusion largely as described previously with modifications [23]. Specifically, after 2 h reperfusion, the hearts were injected with 20 mL of phosphate-buffered saline (PBS) (*w*/*v*, pH 7.4, 37 °C) containing 1% triphenyl tetrazolium chloride through a sidearm of the aortic cannula at the rate of 10 mL/min. Then, the LAD occlusion was reinstated, and 0.1% Evans blue dissolved in KH (*w*/*v*, pH 7.4, 37 °C) was injected at the rate of 0.5 mL/min for 6 min using a syringe pump via a sidearm of the aortic cannula. Then, the hearts were withdrawn from the perfusion system and kept in a freezer at −20 °C for 50–60 min. The frozen hearts were cut into 6 slices and left overnight at 4 °C in 4% formaldehyde solution in PBS. The next day, the heart slices were scanned by an HP scanner. The area at risk (AAR), not stained with Evans blue, and the area of infarct (IS), the white pale zone inside the area at risk (Figure 2), were measured at both sides of each slice using an AlphaEase v5.5 software. The ratio of the total IS to the total AAR of the heart (IS/AAR) was calculated.

### 2.7. Ca^2+^-Induced Mitochondria Swelling Assay

Fast isolation of mitochondria was carried out as we have done previously [24]. All the procedure was performed on ice or at 4 °C. Ventricles were rapidly cut away at the end of preischaemic period, homogenised using a Polytron homogenizer (Kinematica) at 10,000 rpm (two bursts of 5 s followed by one burst of 10 s) in 6 mL of the ISA buffer containing: sucrose 300 mmol/L, EGTA 1 mmol/L, and Tris-HCl 10 mmol/L, pH 7.2. ISA containing 5 mg/mL bovine serum albumin (BSA) was added to the homogenate to the total volume of 30 mL, and the suspension was additionally homogenised for 2 min in a 50 mL glass Potter homogenizer with a Teflon pestle. The homogenate was centrifuged at 2000× *g* for 90 s to remove cell debris (the pellet). The supernatant was then centrifuged at 10,000× *g* for 5 min. The pellet was resuspended in 1 mL of ISA, without EGTA and BSA, and centrifuged at 10,000× *g* for 5 min. The resultant supernatant was poured out and the pellet (the crude mitochondria) were resuspended in 200 μL of the same buffer.

### 2.8. Evaluation of Mitochondria-Bound Hexokinase II (HKII)

Mitochondria were isolated as we have done previously [25,26]. This method allows acquiring pure mitochondria, free from other membranes [25]. In brief, the hearts withdrawn at the end of the preischaemic period were homogenised in ISA buffer using a Polytron and then in ISA buffer containing 5 mg/mL BSA as explained above. The homogenate was centrifuged at 2000× *g* for 90 s and the cell debris was discarded. The supernatant was then centrifuged at 10,000× *g* for 5 min, the pellet was resuspended in 30 mL of ISAPP buffer (ISA supplemented with complete protease inhibitors cocktail (Roche) and phosphatases inhibitors cocktail 1 (Sigma)) and centrifuged at 10,000× *g* for 5 min to obtain a mitochondrial pellet. The pellet was diluted to 6 mL with ISAPP containing 25% (*w*/*v*) Percoll (pH 7.2) and centrifuged at 17,000× *g* for 10 min. The supernatant was carefully pipetted out; the resulting pellet was resuspended in 6 mL of ISAPP and centrifuged again at 7000× *g* for 10 min. The purified mitochondrial pellet was resuspended in 200 μL of ISAPP and stored at −80 °C for later analysis of HKII expression.

### 2.9. Western Blots (Binding of Hexokinase II to Mitochondria)

For determination of HKII binding to mitochondria, frozen mitochondria samples were thawed and the protein concentration was adjusted to a final concentration of 4 mg/mL using BCA protein assay (Fisher Scientific UK Ltd., Loughborough, UK). The samples were diluted 1:1 (*v*/*v*) with the SDS sample buffer (50 mM Tris-HCl, 2 mM EDTA, 12% glycerol, 10% SDS, 5% (*v*/*v*) 2-mercaptoethanol, and 0.01% (*w*/*v*) bromophenol blue). The proteins were separated by 10% SDS-PAGE and processed using Western blotting with antibodies against HKII (Cell Signaling Technology Inc., NEB, Hitchin, UK; diluted 1:1000). Each blot carried samples from control and 8-Br groups of hearts (40 µg per lane). Protein loading was validated using an anti-VDAC antibody (Cell Signaling Technology, Inc.; diluted 1:1000). Western blots were developed using anti-rabbit Ig horseradish peroxidase secondary antibo-Aldrichdy with ECL/ECL+ detection (Amersham Biosciences Ltd, Little Chalfont, UK). The blots were quantified using a digital ChemiDoc imaging system (Bio-Rad, Deeside, UK). AlphaEase v5.5 software was used to analyse band intensity. The data were normalised for protein loading and background was subtracted.

### 2.10. Mitochondrial Permeability Transition Pore (MPTP) Opening

Several methods of measuring MPTP opening are available: Ca^2+^ retention capacity using fluorescence-based analysis [27], [^3^H]DOG entrapment assay [28], measurements of mitochondrial and cytosolic NAD^+^ [29], and others. We measured Ca^2+^-induced mitochondria swelling, which represents a well-established and commonly accepted method of evaluation of MPTP opening in isolated mitochondria [30]. The MPTP opening was measured in de-energised mitochondria at 30 °C using a double-beam spectrophotometer as described previously [28]. The mitochondria swelling was assessed by a decrease in light scattering (at the wavelength of 520 nm) observed after the addition of Ca^2+^ to the suspension of mitochondria. The mitochondria suspension buffer (pH 7.2) contained (mM) 150 KSCN, 20 Mops, 10 Tris, 2 nitrilotriacetic acid, 2 µM A23187 (Sigma-Aldrich, Gillingham, UK), 0.5 µM rotenone (Sigma-Aldrich) and 0.5 µM antimycin A (Sigma-Aldrich). Mitochondria isolated from hearts at the end of 40 min preischaemic period, with or without 5 min perfusion with 8-Br (Figure 1), were diluted to 0.2 mg/mL. Protein concentration was adjusted using a Biuret protein assay. CaCl_2_ was added at 3 mM. The resultant buffered-free Ca^2+^ concentration was 300 µM.

### 2.11. Statistical Analysis

Data are presented as the mean ± SEM. The hearts were randomly distributed between different groups. Statistically significant differences in haemodynamic function between the groups were assessed by one-way ANOVA followed by Fisher’s LSD multiple comparison post hoc test. Statistically significant differences of haemodynamic function of hearts during perfusion with 8-Br vs. pre-treatment values, either before or at the end of regional ischaemia, were evaluated using one-way ANOVA repeated measures followed by Fisher’s LSD multiple comparison post hoc test. Statistically significant differences in arrhythmia score, infarct size, and protein expression in Western blots were assessed using a two-tailed unpaired Student’s *t*-test. The results were considered statistically significant at *p* < 0.05. The post hoc tests were performed if the F-test of variance reached the necessary level of statistical significance. SPSS Statistics software, Version 23 (IBM, Portsmouth, UK), was used for the statistical analysis.

## 3. Results

### 3.1. The Effects of 8-Br Added before Ischemia

#### 3.1.1. Changes in Haemodynamic Function

Heart perfusion with 10 µM 8-Br before the start of regional ischaemia (at the end of preischaemia) resulted in a sharp increase in LVDP, up to 200 ± 18% of the pre-treatment value at the end of the 5 min perfusion with the drug (Figure 3). The following 5 min washout with KH reversed the changes in LVDP, and at the end of preischemic period, this parameter fell below the pre-treatment value. HR was slightly but significantly increased during the heart perfusion with 8-Br and the washout period. Accordingly, RPP rose dramatically during the heart perfusion with 8-Br (up to 230 ± 27%). At the end of the washout period, RPP fell well below the pre-treatment value (to 66 ± 9%; Figure 3).

#### 3.1.2. Changes in Haemodynamic Function during Ischaemia and Reperfusion

In the first series of experiments, the occlusion of the LAD caused a similarly sharp and immediate fall in the LVDP in both the control and 8-Br-Pre groups, to 68 ± 6% and 59 ± 4%, respectively (Figure 3). Then, this parameter started a gradual recovery. At the end of regional ischaemia, LVDP reached 73 ± 9% in the C-Pre and 85 ± 7% in the 8-Br-Pre groups. No significant difference was observed in LVDP between the two groups during regional ischaemia.

During reperfusion, LVDP continued recovering in both groups. In the control group, the increase in LVDP reached its peak (91 ± 9%) at the 40th min. From that point to the end of reperfusion, this parameter did not change significantly. The increase in LVDP in the 8-Br-Pre group was faster and continued longer, until the 60th min of reperfusion, reaching the peak of recovery of 122 ± 8% (*p* < 0.05 vs. control). Afterward, LVDP in 8-Br-Pre hearts continued to be higher compared to the control (Figure 4).

HR remained relatively unchanged during ischaemia and reperfusion in both groups of hearts. Therefore, the dynamics of RPP was similar to LVDP in the C-Pre and 8-Br-Pre groups of hearts (Figure 4).

### 3.2. The Effects of 8-Br Added at the End of Ischemia

#### 3.2.1. Changes in the Haemodynamic Function during 8-Br Treatment

At the onset of reperfusion (at the end of regional ischaemia), heart perfusion with 10 µM 8-Br brought about a fast increase in LVDP, reaching 247 ± 18% of the pre-treatment value at the end of the 5 min perfusion with the drug (Figure 5). During the following 5 min washout period, LVDP returned to the pre-treatment value. HR was insignificantly changed during perfusion of the hearts with 8-Br. RPP increased during the heart perfusion with 8-Br to the maximum of 258 ± 28% and recovered to the pre-treatment level at the end of the washout period (Figure 5).

#### 3.2.2. Changes in Haemodynamic Function during Regional Ischaemia and Reperfusion

In the second series of experiments, changes in the haemodynamic function during regional ischaemia in the C-Post and 8-Br-Post groups followed the pattern of the C-Pre group. From the onset of ischaemia, LVDP fell immediately to 59 ± 5% and 69 ± 9% in C- Post and 8-Br-Post groups, accordingly, and then started gradual recovery until reperfusion in C-Post hearts, or the start of 8-Br perfusion in the 8-Br-Post group. The recovery continued during reperfusion. In C-Post hearts, LVDP reached a maximum of 89 ± 8% on the 50th min of reperfusion. In 8-Br-Post hearts, LVDP recovered faster and continued for longer, similar to the 8-Br-Pre group. This parameter significantly exceeded the values of the C-Post group on the 30th, 60th and 120th min of reperfusion (Figure 6).

HR did not change significantly over the regional ischaemia and reperfusion in both groups of hearts. Therefore, the changes in RPP were similar to those of LVDP. RPP was considerably higher in the 8-Br-Post group compared to control on the 60th and 120th min of reperfusion (Figure 6).

#### 3.2.3. Effect of 8-Br on Ventricular Arrhythmias during I/R

In the first series of experiments, where the hearts were treated with 8-Br prior to regional ischaemia, the number of incidences of VPBs and VT/VF were considerably smaller in 8-Br-Pre hearts. The differences in the number of VT/VF and the arrhythmia score, which combines the number of VPBs and tachyarrhythmias (VT and VF), were significantly lower in 8-Br-Pre hearts compared to the C-Pre group both during ischaemia and reperfusion (Table 1).

In the second series of experiments, during ischaemia, when no heart was subjected to 8-Br treatment, the parameters of ventricular arrhythmias were similar in the C-Post and 8-Br-Post groups. However, perfusion of hearts with 8-Br by the onset of reperfusion significantly reduced the number of incidences of VT/VF and the arrhythmia score during reperfusion (Table 1).

#### 3.2.4. Cardiac Injury

The improved cardiac function recovery and reduced ventricular arrhythmias during regional ischaemia and reperfusion in hearts treated with 8-Br were also accompanied by a significant decrease in cardiac injury assessed by infarct size. Perfusion of hearts with 8-Br by the start of regional ischaemia (8-Br-Pre group) reduced the infarct size-to-area-at-risk ratio (AS/AAR) compared to control (C-Pre group) by 2.5-fold. When the hearts were perfused with 8-Br at the onset of reperfusion (8-Br-Post group), the AS/AAR ratio fell by 3-fold compared to C-Post hearts (Figure 7).

### 3.3. Ca^2+^-Induced Mitochondria Swelling

Perfusion of hearts with 8-Br following the equilibration period with no ischaemia (8-Br-Mito group) significantly increased resistance of the mitochondria to elevated Ca^2+^ concentration manifested in reduced maximal rate and amplitude of mitochondria swelling in these hearts by 2.5-fold (Figure 8).

### 3.4. Hexokinase II Binding to Mitochondria

The relative expression of HKII bound to mitochondria significantly increased in hearts perfused with 8-Br at the end of the equilibration period (Figure 9).

## 4. Discussion

In the experiments on isolated rat hearts, we recently found that the cell-permeable non-selective cAMP analogue 8-Br, which activates the two main cAMP-related signalling pathways, PKA and Epac, can offer a strong cardioprotective effect when applied prior to global ischaemia [16]. These finding, however, are applicable only to cardiac surgery involving aortic cross-clamping, where the onset of an ischaemic event is predictable. Meanwhile, patients often arrive at the hospital with already established AMI caused by regional ischaemia, and the procedure of recanalization of the coronary artery may result in irreversible damage to the myocardium caused by reperfusion [31]. This study shows that brief perfusion of isolated rat hearts with 8-Br is able to protect them against regional ischaemia. Importantly, this cAMP analogue can induce a protective effect when applied not only before ischaemia but also by the onset of reperfusion. Our study also revealed that these effects of 8-Br were likely related to an immediate inhibition of the MPTP mediated by the binding of HKII to the mitochondrial membrane.

### 4.1. 8-Br Protects the Heart against Regional Ischaemia when Applied Either before Ischaemia or by the Onset of Reperfusion

Perfusion of hearts with 8-Br before the start of LAD occlusion induced the protective effect both during regional ischaemia and reperfusion. This was manifested in the improved haemodynamic function, reduced ventricular arrhythmias, and significantly attenuated infarct size (IS/AAR ratio) measured after 2 h of reperfusion.

The cardioprotective effect of the cAMP analogue 8-Br applied by the start of regional ischaemia or reperfusion may seem surprising, since it has been found that stimulation of β-adrenergic receptors (βARs) promotes cardiac arrhythmias through increased levels of cAMP, activation of PKA and Epac [32,33]. Thus, it has been shown that the cardiotonic drug milrinone, which inhibits the phosphodiesterase 3 (PDE3) family to improve cardiac function by elevating cAMP, promotes sudden cardiac death due to arrhythmias [34]. PDE4 downregulation has been shown to enhance aberrant spontaneous pro-arrhythmic release of Ca^2+^ [35,36]. Bobin et al. [37] have also found that β-adrenergic stimulation by inhibition of PDE4 incurs positive inotropic effects, but these effects are associated with the increased sarcoplasmic reticulum Ca^2+^ load and leak, resulting in the increased occurrence of proarrhythmic spontaneous Ca^2+^ waves via cAMP-dependent PKA and Epac pathways. Fazal et al. have shown that in failing human hearts, characterised by chronic stimulation of β-adrenergic receptors, the expression of Epac1 is increased [38]. Further, the authors used mice with genetic deletion of *Epac1* to show that chronic inhibition of Epac1 activity is cardioprotective. The deleterious effects of Epac1, particularly mitochondrial Epac1, were explained by Ca^2+^ overload and opening of the MPTP. Laudette et al. in their review have also discussed the deleterious effects of Epac in cardiovascular and pulmonary systems, and these effects were associated with chronic stimulation of cAMP signalling pathways in conditions like myocardial remodelling, heart failure, and chronic obstructive pulmonary disease. However, the modality of the effects of cAMP signalling depends on several factors such as the timing, duration and dose of the treatment as well as the balance between activation of Epac- and PKA-related pathways [16]. In particular conditions, an increased level of cAMP is cardioprotective, and one of the manifestations of this effect can be a reduction in cardiac arrhythmias. It is well known that ischaemic preconditioning (IP) [39,40,41] and temperature preconditioning [42] prevent cardiac arrhythmias and this effect is associated with elevated cAMP levels and PKA activation [19,43,44]. Furthermore, previous studies of Lochner et al. [9] showed that activation of PKA by stimulation of βAR triggers a protective effect similar to that of IP.

One of the possible explanations of the antiarrhythmic effects of IP is attenuation of free cytosolic Ca^2+^ during prolonged ischemia due to the reduced ryanodine-dependent release of Ca^2+^ from the sarcoplasmic reticulum [45], and by inhibited Ca^2+^ influx, efflux or redistribution [46]. These effects are, in fact, opposite to the proarrhythmic effects of cAMP signalling described above. This seeming controversy can be explained by the timing and short duration of the cardioprotective intervention that stimulates cAMP signalling. Apparently, a short (5 min) washout period, or its absence, determines the impact of the elevated cAMP on the myocardial injury during I/R. Thus, Salie et al. [47] have recently shown in experiments on isolated rat heart that during the 5 min washout period after IP induced by a 5 min global ischaemia, activation of the cell survival kinases, such as p38MAPK, JNKp54/p46, Akt and extracellular signal-regulated kinase (ERK), takes place that might determine the cardioprotective effect of PKA. Others have suggested that the cAMP-dependent antiarrhythmic effect of IP is mediated by enhancing vagal influences [41]. The latter hypothesis is consistent with the fact that LVDP and RPP at the end of the washout period in the 8-Br-Pre group of hearts fell below the pre-treatment values. Also, HR in 8-Br-Pre hearts was significantly elevated over the pre-treatment level on the 4th min of the washout period but fell to the pre-treatment value on the 5th min (Figure 3). It can be suggested that the brief perfusion of the heart with 8-Br leads to activation of the intramural parasympathetic ganglia, which remain intact in a Langendorff-perfused rat heart, resulting in the release of acetylcholine in the ventricles and induction of protection in the ventricles by activation of muscarinic receptors, as it has been found for hearts subjected to IP [48]. In 8-Br-Post hearts, haemodynamic function returned to the pre-treatment values but did not fall further. This, however, does not exclude the possibility of intramural parasympathetic ganglia activation, because the haemodynamic effects of this activation could be hidden by the βAR activation taking place during ischaemia [49]. Earlier studies have shown that cytoprotective effects of stimulation of these receptors could be mediated by increased activities of anti-apoptotic enzymes Bcl-2, ERK and superoxide dismutase, decreasing the pro-apoptotic mediators Fas and p38 [50]. Other possible mechanisms of cardioprotection associated with the cAMP signalling may involve the opioidergic system [51], prostanoids [52], nitric oxide [53], PKCε [16] or adenosine [54]. However, further studies are needed to investigate the involvement of these suggested signalling molecules and pathways in the cardioprotective effect of 8-Br.

We [18] and others [55,56,57] have found that MPTP opening is a core factor of reperfusion heart injury. Opening of this non-selective pore at the onset of reperfusion for a relatively long period of time leads to breakdown of the electron transport chain, ROS production, rupture of mitochondrial membranes, release, and activation of the proapoptotic, necrotic and proinflammatory factors, and ultimately, irreversible damage to the myocardium. Therefore, inhibition of MPTP represents an important mechanism of cardioprotection [58,59], and our finding of the ability of 8-Br to inhibit this pore is one of the major results of this study.

### 4.2. Perfusion of Hearts with 8-Br Inhibits MPTP

The results of this work show that perfusion of hearts with 8-Br led to an immediate increase in binding of HKII to mitochondria and inhibition of Ca^2+^-induced mitochondria swelling, a well-established marker of the resistance of MPTP to Ca^2+^ overload [60]. The binding of HKII to the outer mitochondrial membrane (OMM) has been linked to inhibition of MPTP and anti-apoptotic effects [61,62], and it is negatively correlated to the extent of ischaemia and infarct size [26]. Increased binding of HKII can be associated with decreasing levels of glycogen, which results in a reduction in acidification and tissue glucose-6-phosphate (G6P) levels that hamper HKII dissociation from mitochondria [26]. Although we did not measure the levels of glycogen and G6P in our study, the increase in LVDP and RPP by 2–2.5-fold during the heart perfusion with 8-Br, either before or during the regional ischaemia, might lead to glycogen depletion in the myocardium and promote binding of HKII to the outer mitochondrial membrane. These results are consistent with our previous study on isolated rat hearts showing that isoproterenol (or isoprenaline), which activates cAMP signalling through stimulation of βARs, reduces glycogen content in the myocardium and promotes binding of HKII to mitochondria [11]. Of note, this work showed that stimulation of βARs by isoproterenol only induced a mild protective effect. Our recent study has revealed that the optimal cardioprotective effect of the elevated levels of cAMP signalling can be achieved as a result of simultaneous activation of its two main downstream targets, PKA and Epac [16]. Meanwhile, Purves et al. found that PKA is more sensitive to small elevations in cAMP, while Epac needs higher levels of cAMP for its activation [63]. This is consistent with the finding that PKA is more sensitive to cAMP than Epac [64]. In contrast, Dao et al. found that cAMP displays a similar affinity to PKA I and Epac1, and both PKA and Epac can react to similar physiological concentrations of cAMP [65]. However, which of these proteins are preferentially activated by cAMP depends not only on the affinity but also on the proximity of PKA and Epac to cellular compartments with elevated levels of cAMP (e.g., βAR) [66]. It can be speculated that βAR stimulation by isoproterenol results in preferential activation of PKA, either due to the different sensitivities or closer proximity to βAR, whilst direct administration of the cell-permeable cAMP analogue 8-Br results in a more equal activation of these two enzymes. It should be also taken into account that isoproterenol is a non-selective βAR agonist which can activate any type of βARs. However, in contrast to βAR1, which acts through translocation of G_s_ protein and activation of the adenylate cyclase with the formation of cAMP [67], βAR2 may activate G_i_ protein, and, thus, attenuate the effects of cAMP via activation of guanylyl cyclase [68]. Thus, the difference in the cardioprotective efficacy between 8-Br and isoproterenol could be associated with either an insufficient activation of Epac by the βAR stimulation with isoproterenol or/and with a possible inhibition of the cAMP signalling through activation of G_i_ protein when stimulation of the βARs by isoproterenol is relatively high.

The binding of HKII to the OMM plays both metabolic and membrane-stabilising roles. Thus, it has been shown that the increased cytochrome c release, ROS production, and infarct size on reperfusion in hearts with depleted mitochondria-bound HKII was associated with the opening of the MPTP, and increased permeabilization of the OMM with degradation of Bcl-x_L_ [26]. Others have found that HKII bound to OMM prevents binding of the pro-apoptotic enzyme BAX to mitochondria, which also prevents cytochrome c release from the mitochondria [69]. The enzymatic effects of increased HKII binding to OMM are manifested in the prevention of acidosis via the enhanced coupling of glycolysis and glucose oxidation and inhibition of fatty acid oxidation. This metabolic function of the mitochondria-bound HKII also results in inhibition of ROS production [70,71]. Therefore, the enzyme activity of HKII in the membrane-bound state has its own protective value, which needs to be further investigated in relation to the cardioprotective effects of 8-Br. Using an inhibitor of HKII, such as 3-bromo pyruvate, could help to differentiate the protective role of HKII associated with the enzymatic activity from its membrane-stabilising function. However, for these experiments, the enzymatic activity needs to be inhibited in the membrane-bound but not in the dissolved state of HKII.

It is important to stress that we perfused the hearts with 8-Br at the end of regional ischaemia in order to ensure that the drug affects the MPTP at the very beginning of reperfusion because the opening of this pore is observed during the first minutes of reperfusion [72]. Importantly, we found that 8-Br immediately promotes binding of HKII to mitochondria and inhibits MPTP (reduces Ca^2+^-induced mitochondria swelling), which is critical for salvaging the myocardium in these conditions. Considering the importance of MPTP opening for reperfusion injury, we may suggest that this effect is a key factor of the cardioprotective effect of 8-Br applied at the onset of reperfusion. However, a recent work of Pereira et al. has demonstrated that in vitro dissociation of HKII alone does not fully replicate the effects of global ischaemia on mitochondrial function [73]. The authors concluded that in vivo dissociation of HKII regulates mitochondrial function at the end of ischaemia indirectly, probably by interaction with mitochondrial fission proteins such as Drp1 and Drp2. They hypothesised that the changes in cristae structure and mitochondrial morphology occurred due to this interaction destabilised the integration between the outer and inner membrane, resulting in increased cytochrome c release and sensitising MPTP to Ca^2+^ overload.

It should be noted that our studies were performed on healthy mitochondria. In diseased conditions, the reaction of mitochondria to 8-Br treatment could differ from healthy mitochondria. In the future, the effect of 8-Br on the binding of HKII to mitochondria and MPTP opening can be tested in more clinically relevant diseased models (e.g., heart failure following coronary artery occlusion) where the isolated mitochondria will come from a diseased heart. Also, these measurements need to be accompanied by evaluation of cytochrome c release from mitochondria and activation of caspase cascade leading to apoptosis. However, we would like to point out that the protective effect manifested in the reduction of cardiac arrhythmias, infarct size, and the improved haemodynamic function was observed in hearts treated with 8-Br not only before but also at the end of regional ischaemia, indicating the potential of 8-Br to protect the heart in a pathological state. The distribution of 8-Br over the ischaemic and non-ischaemic areas during the coronary artery occlusion and the reaction of these mitochondria to the treatment with 8-Br need to be studied in future experiments.

It needs to be taken into account that the results obtained in experiments on small rodents, including those on the pathological models, cannot be directly extrapolated to humans [74]. To advance the translation of these results to clinical practice, the protective effects of 8-Br need to be tested on a large animal model, such as a pig, in which morphological and physiological characteristics of the cardiovascular system are closely matched to humans [75]. Over the years, we have developed expertise in porcine models of cardiac surgery with cardio-pulmonary bypass and cardioplegic arrest [76,77]. This experience can be employed should the experiments on pathological in vivo and ex vivo models in rats confirm the potent cardioprotective effect of 8-Br.

Taken together, the results of this study show for the first time that the cell-permeable cAMP analogue 8-Br is able to protect the heart against regional ischaemia/reperfusion injury when applied either before ischaemia or by the onset of reperfusion. These findings widen the clinical relevance of 8-Br as a potential drug for the prevention of ischaemia/reperfusion injury occurring during the coronary recanalization procedure in AMI patients as well as in heart surgery involving aortic cross-clumping. Immediate binding of HKII to mitochondria and inhibition of MPTP are likely mechanisms underlying this protective effect.

## Figures and Tables

**Figure 1 cells-10-01223-f001:**
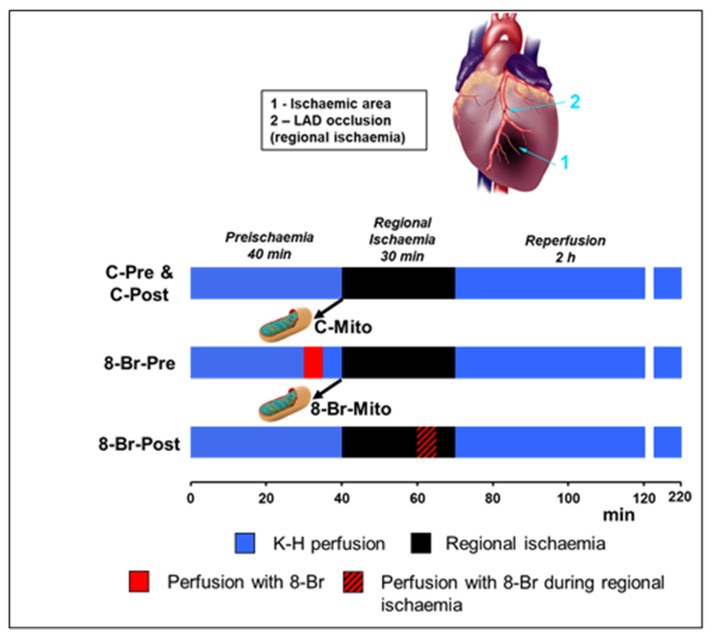
Experimental protocol. Upper panel—a schematic representation of the regional ischaemia induced by the LAD occlusion. Lower panel—protocols of heart perfusion and ventricular sample collection. 8-Br was perfused for 5 min at the concentration of 10 μM followed by a 5 min washout period. Hearts of C-Mito and 8-Br-Mito groups (*n* = 5 and 7, respectively) were not subjected to ischaemia and used for isolation of mitochondria. C-Pre and C-Post—control groups of hearts for the First and Second series of experiments (*n* = 7 and 6, respectively). 8-Br-Pre and 8-Br-Post—groups of hearts treated with 8-Br either before regional ischaemia (first series of experiments) or at the end of 30 min regional ischaemia (second series of experiments).

**Figure 2 cells-10-01223-f002:**
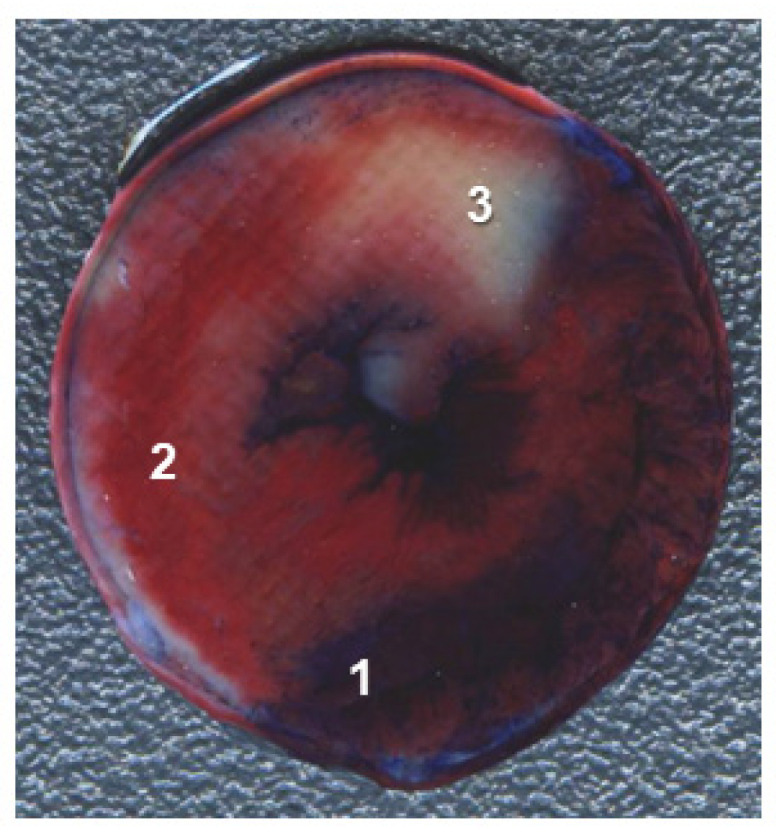
A representative image of a heart slice. 1—Non-ischaemic area stained with Evan Blue; 2—non-infarcted area at risk; 3—infarcted area.

**Figure 3 cells-10-01223-f003:**
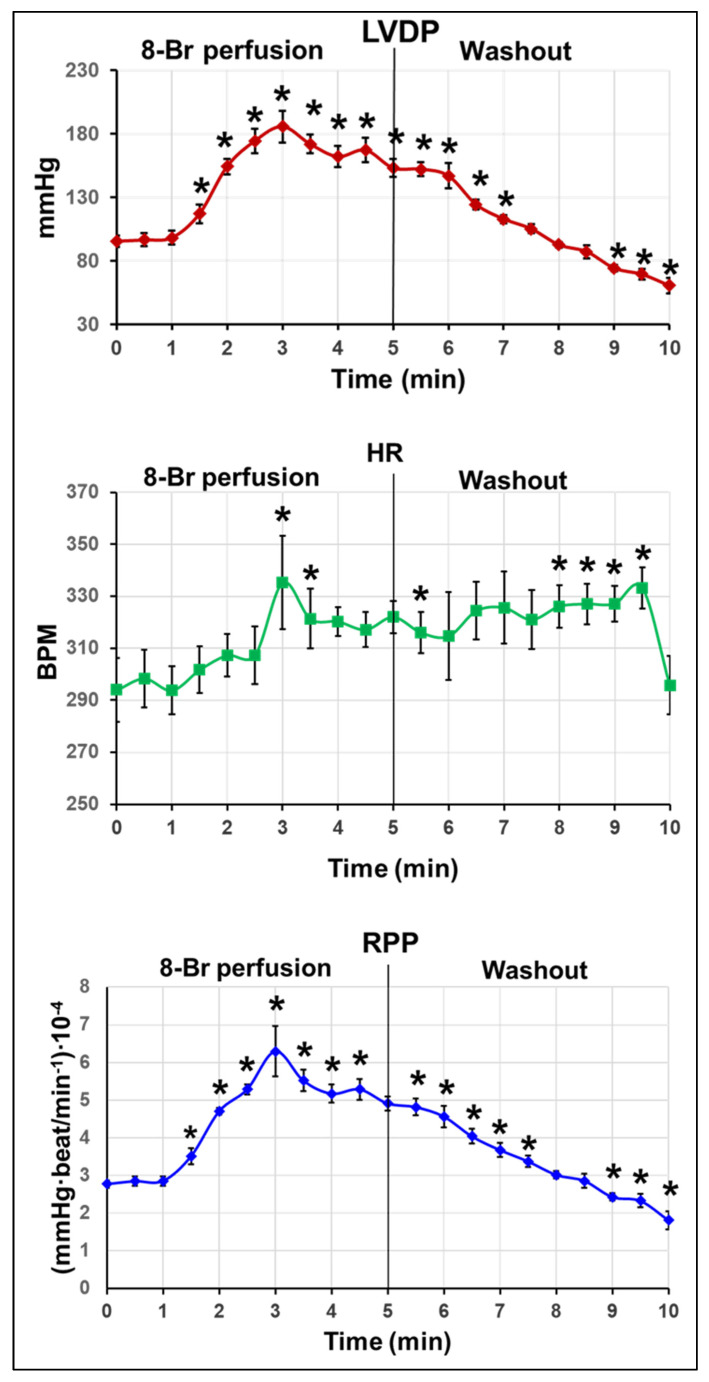
Haemodynamic function of hearts treated with 8-Br before regional ischaemia. Changes in LVDP, HR, and RPP are presented as the mean ± SEM. * *p* < 0.05 vs. the pre-treatment value (at 0 min).

**Figure 4 cells-10-01223-f004:**
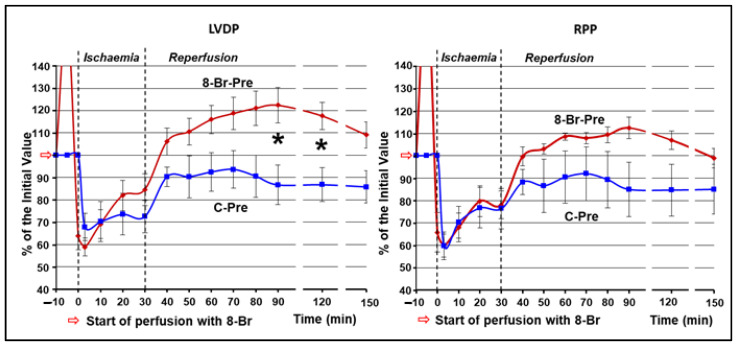
Changes in the RPP and LVDP of the hearts during regional ischaemia and reperfusion in the first series of experiments. In this series of experiments, the hearts were treated with 8-Br before the regional ischaemia. * Statistically significant difference between the C-Pre and 8-Br-Pre groups of hearts (*p* < 0.05).

**Figure 5 cells-10-01223-f005:**
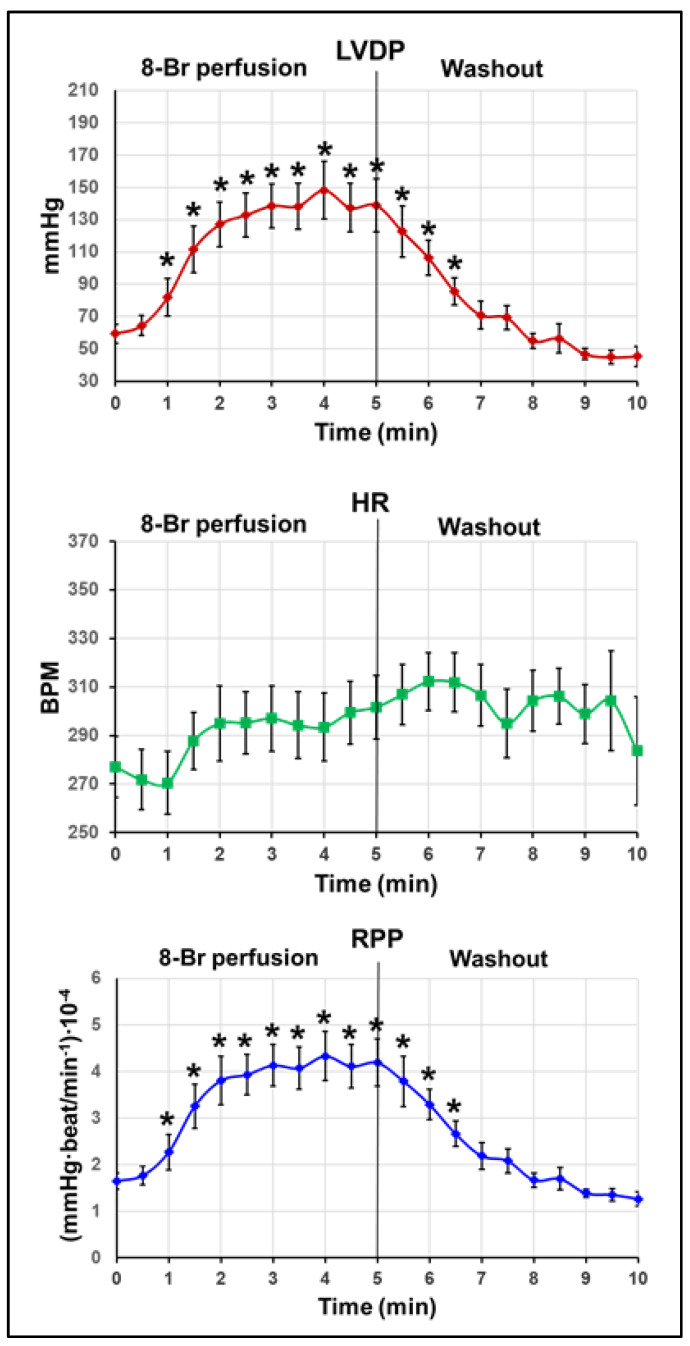
Haemodynamic function of hearts treated with 8-Br before reperfusion. Changes in LVDP, HR, and RPP are presented as the mean ± SEM. * *p* < 0.05 vs. the pre-treatment value (at 0 min).

**Figure 6 cells-10-01223-f006:**
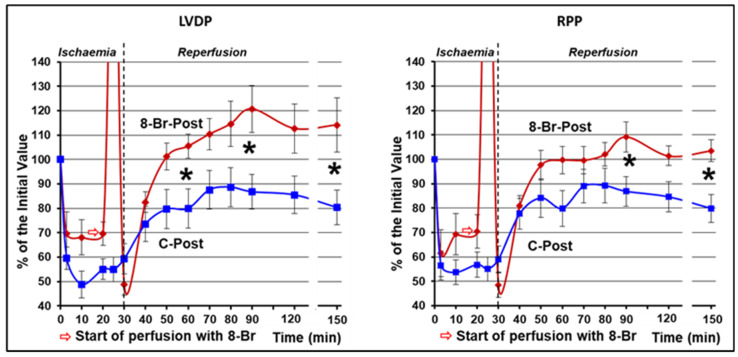
Changes in RPP and LVDP of the hearts during regional ischaemia and reperfusion in the second series of experiments. In this series of experiments, the hearts were treated with 8-Br by the onset of reperfusion. * Statistically significant difference between the C-Pre and 8-Br-Pre groups of hearts (*p* < 0.05).

**Figure 7 cells-10-01223-f007:**
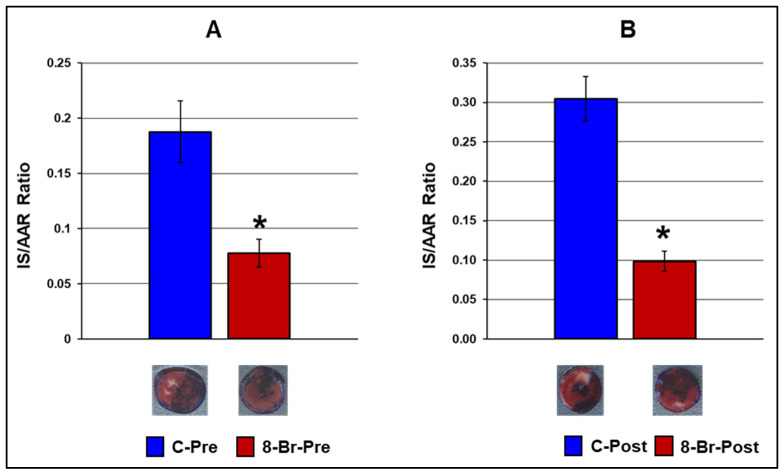
Infarct size in hearts of the first and second series of experiments. (**A**) First series of experiments. The hearts were treated with 8 Br prior to regional ischaemia (8-Br-Pre group) or no treatment with 8-Br (C-Pre group). (**B**) Second series of experiments. The hearts were treated with 8-Br at the end of regional ischaemia (8-Br-Post group) or received no treatment (C-Pre group). The data are presented as the mean ± SEM. * *p* < 0.05 vs. 8-Br-Pre or 8-Br-Post group, respectively. Representative images of the stained heart slices of each group are shown under the corresponding bars. The red area, not stained with Evance blue, is the area at risk. The dark-blue area is non-ischaemic myocardium, and the white, pale zone inside the area at risk is the infarcted (necrotic) area.

**Figure 8 cells-10-01223-f008:**
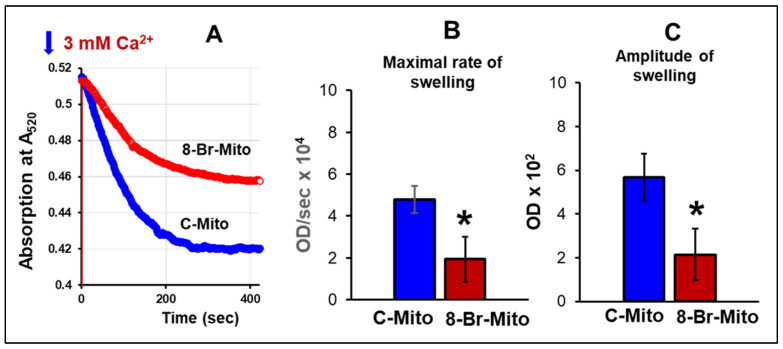
Effect of heart perfusion with 8-Br on Ca2+-induced mitochondrial swelling. (**A**) Representative traces of the increase in light scattering (fall of light absorption (arbitrary units) at A520) due to the mitochondria swelling in the presence of 3 mM Ca2+. The mitochondria swelling was measured immediately after mitochondria isolation following heart perfusion with 10 μM 8-Br (8-Br-Mito group) or without any additional drug (C-Mito group) during preischaemia. (**B**) The maximal rate of swelling. (**C**) Amplitude of swelling. The data are presented as the mean ± SEM. * *p* < 0.05 vs. the C-Mito group of hearts.

**Figure 9 cells-10-01223-f009:**
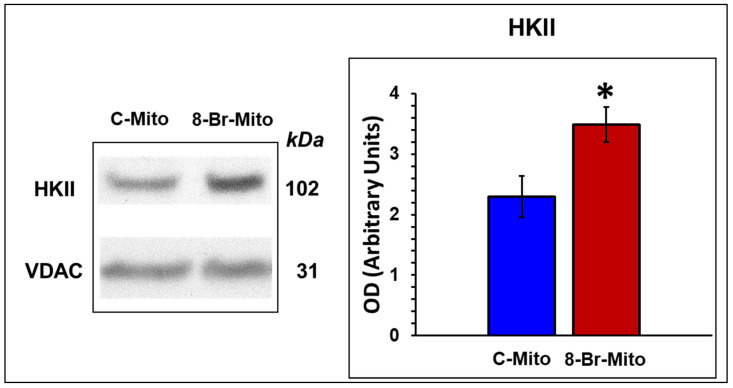
Effect of heart perfusion with 8-Br on Hexokinase II (HKII) expression bound to mitochondria. The expression of HKII bound to mitochondria was assessed using Western blotting. Groups of hearts not subjected to ischaemia: C-Mito, control; 8-Br-Mito, perfused with 10 μM 8-Br. The optical density (OD) of HKII was normalised for the OD of VDAC and presented as mean ± SEM. * *p* < 0.05 vs. the C-Mito group of hearts.

**Table 1 cells-10-01223-t001:** Effects of 8-Br on ventricular arrhythmias during regional ischaemia and reperfusion.

Parameters		C-Pre (*n* = 7)	8-Br-Pre (*n* = 5)	C-Post (*n* = 6)	8-Br-Post (*n* = 7)
**Regional Ischaemia**	**Number of VPBs**	830 ± 194	401 ± 126	738 ± 193	856 ± 267
	**Number of VT/VF**	38 ± 15	3 ± 1 *	77 ± 27	39 ± 15
	**Arrhythmia Score**	6.8 ± 0.2	5.0 ± 0.8 *	6.8 ± 0.2	6.7 ± 0.3
**Reperfusion**	**Number of VPBs**	409 ± 117	160 ± 62	211 ± 77	249 ± 62
	**Number of VT/VF**	10 ± 2	3 ± 1 *	9 ± 2	3 ± 1 *
	**Arrhythmia Score**	6.3 ± 0.2	4.8 ± 0.4*	6.4 ± 0.3	4.7 ± 0.6 *

8-Br-Pre—a group of hearts treated with 10 μM 8-Br at the end of preischaemia. 8-Br-Post—a group of hearts treated with 10 μM 8-Br at the end of regional ischaemia. C-Pre and C-Post—control groups of hearts corresponding to 8-Br-Pre and 8-Br-Post. The data are presented as the mean ± SEM. * *p* < 0.05 vs. control (C-Pre or C-Post, respectively).

## Data Availability

All relevant data are included within the manuscript. The raw data are available on request from the corresponding author.

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
