# Peer review of "Preconditioning or Postconditioning with 8-Br-cAMP-AM Protects the Heart against Regional Ischemia and Reperfusion: A Role for Mitochondrial Permeability Transition"

_cells, 2021, doi:10.3390/cells10051223_

Round 1
Reviewer 1 Report
It is an interesting paper on the cardioprotective effect of a cAMP analog, 8-Br-cAMP-AM when applied prior to regional ischaemia or at the onset of reperfusion following regional ischaemia. The authors determined the effect of the drug on infarct size, haemodynamic function and ventricular arrhythmias during regional ischaemia and reperfusion. Mechanistically, 8-Br leads to an increase in binding of hexokinase II to mitochondria and inhibition of Ca2+-induced mitochondria swelling.
The MS is a follow up of a previous study performed by the same group (Khaliulin I et al.. Br J Pharmacol. 2017 Mar;174(6):438-453). Overall, the study background is clear and the experimental approach is appropriate to provide clear results and conclusions.
I have only minor comments:
- The authors should discuss the functional role of Epac proteins in cardiac ischemia as shown in Fazal et al., Circ Res. 2017, Laudette et al., Cardiovasc. Res 2019.
- 360: The authors referred to [29] for the effect of PKA and Epac on cardiac arrhythmias. This is not the exact reference. Please cite: Pereira et al. J. Physiol. 2007, Circulation 2013
- 361: “increases” and not “increased”
- Purves et al. should be added in the ref. list.
- 443-446. It has been described that Epac1 and PKA have similar cAMP affinity (Dao KK, et al., J Biol Chem. 2006 Jul 28;281(30):21500-21511. The authors should also consider the duration of B-AR stimulation. Acute B-AR activation preferentially activates PKA while prolonged B-AR stimulation switches on Epac/CaMKII pathway (Laudette M et al., 2019, 2021). Please discuss these points.
Author Response
Reviewer 1
- The authors should discuss the functional role of Epac proteins in cardiac ischemia as shown in Fazal et al., Circ Res. 2017, Laudette et al., Cardiovasc. Res 2019.
Response:
We thank the reviewer for this suggestion. We have now added the information about these publications to the Discussion (Lines 376-385). As we point out in our manuscript, the modality of the effects of cAMP signalling depends on several factors, including the dose, timing, and duration of the activation of this pathway. In our experiments, we used brief, transient perfusion of hearts with the activator of cAMP signalling 8-Br-cAMP-AM, which brought about a potent cardioprotective effect. The mentioned works of Frank Lezoualc’h’s group [1,2] discuss the effect of chronic activation of this pathway which leads to detrimental effects. Thus, Fazal et al. have shown that in failing human hearts, characterized by chronic stimulation of β-adrenergic receptors (βAR), the expression of Epac1 is increased. Further, the authors used mice with genetic deletion of Epac1 to show that chronic inhibition of Epac1 activity is cardioprotective. The deleterious effects of Epac1, particularly mitochondrial Epac1, were explained by Ca2+ overload and opening of the MPTP. Laudette et al. in their review also discuss the deleterious effects of Epac in cardiovascular systems, and these effects were associated with chronic stimulation of cAMP signalling pathways in conditions like myocardial remodeling, heart failure, and chronic obstructive pulmonary disease.
- 360: The authors referred to [29] for the effect of PKA and Epac on cardiac arrhythmias. This is not the exact reference. Please cite: Pereira et al. J. Physiol. 2007, Circulation 2013
Response:
We have replaced the reference [29] with the two references of L. Fereira. These are citations [33, 34] now (Line 368).
- 361: “increases” and not “increased”
Response:
We have changed “increased” to “increases” (Line 367).
- Purves et al. should be added in the ref. list.
Response:
We have added the reference of Purves et al (ref. [67], Line 456).
- 443-446. It has been described that Epac1 and PKA have similar cAMP affinity (Dao KK, et al., J Biol Chem. 2006 Jul 28;281(30):21500-21511. The authors should also consider the duration of B-AR stimulation. Acute B-AR activation preferentially activates PKA while prolonged B-AR stimulation switches on Epac/CaMKII pathway (Laudette M et al., 2019, 2021). Please discuss these points.
Response:
This is an interesting addition to our discussion. Indeed, in contrast to Purves et al. [3], Dao et al. have found that cAMP displays a similar affinity to PKA I and Epac1, and both PKA and Epac can react to similar physiological concentrations of cAMP [4]. However, which of these proteins are preferentially activated by cAMP depends not only on the affinity but also on the proximity of PKA and Epac to cellular compartments with elevated levels of cAMP (e.g. βAR) [5]. It can be suggested that PKA is preferentially activated by βAR than Epac either due to higher sensitivity or closer proximity to the cellular compartments with elevated cAMP. We added this discussion to the manuscript (Lines 457-463).
Regarding the second part of this comment, we were unable to find the publication of al. showing that acute stimulation of βAR preferentially activates PKA while sustained stimulation of these receptors switches on the Epac pathway. However, some publications of this group confirmed that the protective effects of cAMP are associated with acute βAR stimulation, while chronic stimulation of βAR leads to heart pathology [6,7]. This exactly matches our discussion of the first comment.
References
- Fazal, L.; Laudette, M.; Paula-Gomes, S.; Pons, S.; Conte, C.; Tortosa, F.; Sicard, P.; Sainte-Marie, Y.; Bisserier, M.; Lairez, O. Multifunctional mitochondrial Epac1 controls myocardial cell death. Circ Res 2017, 120, 645-657.
- Laudette, M.; Zuo, H.; Lezoualc’h, F.; Schmidt, M. Epac function and cAMP scaffolds in the heart and lung. J Cardiovasc Dev Dis 2018, 5, 9.
- Purves, G.I.; Kamishima, T.; Davies, L.M.; Quayle, J.M.; Dart, C. Exchange protein activated by cAMP (Epac) mediates cAMP‐dependent but protein kinase A‐insensitive modulation of vascular ATP‐sensitive potassium channels. J Physiol 2009, 587, 3639-3650.
- Dao, K.K.; Teigen, K.; Kopperud, R.; Hodneland, E.; Schwede, F.; Christensen, A.E.; Martinez, A.; Døskeland, S.O. Epac1 and cAMP-dependent protein kinase holoenzyme have similar cAMP affinity, but their cAMP domains have distinct structural features and cyclic nucleotide recognition. J Biol Chem 2006, 281, 21500-21511.
- Baillie, G.S.; Houslay, M.D. Arrestin times for compartmentalised cAMP signalling and phosphodiesterase-4 enzymes. Curr Opin Cell Biol 2005, 17, 129-134.
- Laudette, M.; Formoso, K.; Lezoualc’h, F. GRKs and Epac1 Interaction in Cardiac Remodeling and Heart Failure. Cells 2021, 10, 154.
- Lezoualc’h, F.; Fazal, L.; Laudette, M.; Conte, C. Cyclic AMP sensor EPAC proteins and their role in cardiovascular function and disease. Circ Res 2016, 118, 881-897.
Reviewer 2 Report
This manuscript describes the effects of 8-Bromo-CAMP-AM on regional ischemia and reperfusion paradigm measured in rats (2 month old wistar rats). The authors prvided a sophistacted experimental setting to compare the infusion of 8-Bromo-cAMP-AM on global ischemia compared to regional ischemia. The data presented implicate that (short-term) perfusion of 8-Bromo-cAMP-AM protect against heart damage induced by ligation of LAD. Evidence is provided that 8-Bromo-cAMP-AM reduced mitochondria swelling and restores binding of hexokinase II to mitochondria.
The provided data are robust but the authors should comment on the following points to further improve the current manuscript.
1) The authors provide evidence that 8-Bromo-cAMP-AM reduced swelling of mitochondria. The provided data are convincing. However, the mitochondria used were from hearts not subjected to ischemia. The authors should provide more details. Does this match the patient situation? Most likely "damaged" mitochondria (for example due to ischemic damage) might react differently to 8-Bromo-cAMP-AM. What is the rational to isolate mitochondria from non-ischemic heart only?
2) The binding of Hexokinase II to mitochondria is increased by 8-Bromo-cAMP-AM. Please comment on the rational to enroll for the studies "healthy" mitochondria only? Particularly local regional ischemia might change the sensitivity to 8-Bromo-cAMP-AM profoundly?
3) The authors implicate that the protective effects of 8-Bromo-cAMP-AM on ischemic damage of the heart might be due reduction in mitochondria swelling process linked to the binding of hexokinase II. To further prove this conclusion would it be possible to study the role of hexokinase II by using inhibitors such as 3-bromo pyruvate. The authors should comment if binding of hexokinase II is sufficient of if its activity is required.
4) The authors should comment on the stability of 8-Bromo-cAMP-AM. If used in daily practice (in patient setting) what about the metabolic stability of this cAMP derivative?
Author Response
Reviewer 2
1) The authors provide evidence that 8-Bromo-cAMP-AM reduced swelling of mitochondria. The provided data are convincing. However, the mitochondria used were from hearts not subjected to ischemia. The authors should provide more details. Does this match the patient situation? Most likely "damaged" mitochondria (for example due to ischemic damage) might react differently to 8-Bromo-cAMP-AM. What is the rational to isolate mitochondria from non-ischemic heart only?
Response:
Unlike patients, the model we used was that of a normal heart. We followed a standard experimental protocol in our investigation by testing the interventions and optimizing the conditions using normal rodent hearts first. This can then be tested in more clinically relevant diseased models (e.g. heart failure following coronary artery occlusion) where the isolated mitochondria will come from a diseased heart. Therefore, we agree with the reviewer that during ischaemia, mitochondria may react differently to the treatment, and the reaction of mitochondria in the ischaemic zone needs to be investigated in future experiments. We added these points to the Discussion (Lines 506-511).
2) The binding of Hexokinase II to mitochondria is increased by 8-Bromo-cAMP-AM. Please comment on the rational to enroll for the studies "healthy" mitochondria only? Particularly local regional ischemia might change the sensitivity to 8-Bromo-cAMP-AM profoundly?
Response:
As we explained in our response to the previous question, we used healthy hearts as a standard model for an initial study. We agree with the reviewer that mitochondria in the ischaemic zone may react differently to 8-Br-cAMP-AM (8-Br), including the binding of Hexokinase II, and this needs to be investigated in future experiments on a model of damaged myocardium. These thoughts are added to the Discussion (Lines 506-511).
Meanwhile, we would like to point out that the protective effect manifested in the reduction of cardiac arrhythmias, infarct size, and the improved haemodynamic function was observed in hearts treated with 8-Br not only when added before, but also at the end of regional ischaemia. The distribution of 8-Br over the ischaemic and non-ischaemic areas during the coronary artery occlusion and the reaction of these mitochondria to the treatment with 8-Br need to be studied in future experiments. We added these thoughts to the Discussion (Lines 513-518).
3) The authors implicate that the protective effects of 8-Bromo-cAMP-AM on ischemic damage of the heart might be due reduction in mitochondria swelling process linked to the binding of hexokinase II. To further prove this conclusion would it be possible to study the role of hexokinase II by using inhibitors such as 3-bromo pyruvate. The authors should comment if binding of hexokinase II is sufficient of if its activity is required.
Response:
Hexokinase II (HKII) exerts its enzyme activity, the ATP-dependent phosphorylation of glucose to glucose-6-phosphate (G6P) both in membrane-bound and dissolved state. The binding of HKII to the outer mitochondrial membrane (OMM) plays both metabolic and membrane-stabilizing roles. Thus, it has been shown that the extent of mitochondrial HKII dissociation during ischemia in isolated rat hearts correlates with cytochrome c release, reactive oxygen species (ROS) production, and infarct size on reperfusion. This effect was associated with the opening of the MPTP, and increased permeabilization of the OMM with degradation of Bcl-xL [1]. Others have found that HKII bound to OMM prevents binding of the pro-apoptotic enzyme BAX to mitochondria, which also prevents cytochrome c release from the mitochondria [2]. Increasing HKII binding to OMM also prevents acidosis via the enhanced coupling of glycolysis and glucose oxidation and inhibits oxidation of fatty acid. This metabolic function of the mitochondria-bound HKII also leads to inhibition of ROS production [3,4]. Therefore, the enzyme activity of HKII in the membrane-bound state has its own protective value which needs further investigation. The effects of binding of HKII to OMM have been studied by changing the concentration of G6P. Increased concentration of G6P results in dissociation of HKII from the OMM. Using an inhibitor of HKII could help to differentiate the protective role of HKII associated with the enzymatic activity from its membrane-stabilizing function. However, for these experiments, a technical issue needs to be solved: the enzymatic activity needs to be inhibited in the membrane-bound but not in the dissolved state of . We added this information to the Discussion (Lines 474-489).
References
- Pasdois, P.; Parker, J.E.; Halestrap, A.P. Extent of mitochondrial hexokinase II dissociation during ischemia correlates with mitochondrial cytochrome c release, reactive oxygen species production, and infarct size on reperfusion. J Am Heart Assoc 2012, 2, e005645.
- Pastorino, J.G.; Shulga, N.; Hoek, J.B. Mitochondrial binding of hexokinase II inhibits Bax-induced cytochrome c release and apoptosis. Journal of Biological Chemistry 2002, 277, 7610-7618.
- da-Silva, W.S.; Gómez-Puyou, A.; de Gómez-Puyou, M.T.; Moreno-Sanchez, R.; De Felice, F.G.; de Meis, L.; Oliveira, M.F.; Galina, A. Mitochondrial bound hexokinase activity as a preventive antioxidant defense: steady-state ADP formation as a regulatory mechanism of membrane potential and reactive oxygen species generation in mitochondria. J Biol Chem 2004, 279, 39846-39855.
- Roberts, D.; Miyamoto, S. Hexokinase II integrates energy metabolism and cellular protection: Akting on mitochondria and TORCing to autophagy. Cell Death Differ 2015, 22, 248-257.
Reviewer 3 Report
This is an interesting study which investigates the ability of 8-Br-cAMP-AM treatment before/at the end of ischemia to offer protection to the rat heart attenuating ventricular arrhythmias, improving haemodynamic function and reducing infarct size.
I only have a couple of comments that I would like the authors to address:
1. I wasn`t really sure whether sufficient evidence was presented to categorically confirm that there was mitochondrial permeability transition pore opening. No assessment of cytochrome c release or caspase activation were undertaken and the authors should indicate this in the manuscript.
2. It has been suggested in the study by Pereira et al (2020) in the PLOS ONE journal that a number of other factors in addition to HKII binding to the mitochondrial membrane protect against mitochondrial permeability transition pore opening. The authors should include this discussion point in the paper.
3. This is a study undertaken in rats and it is uncertain whether these results would translate into human studies. Some discussion about this fact would be pertinent in the manuscript.
Author Response
Reviewer 3
- I wasn`t really sure whether sufficient evidence was presented to categorically confirm that there was mitochondrial permeability transition pore opening. No assessment of cytochrome c release or caspase activation were undertaken and the authors should indicate this in the manuscript.
Response:
Several methods of measuring MPTP opening are available: Ca2+ retention capacity using fluorescence-based analysis [1], [3H]DOG entrapment assay [2], measurements of mitochondrial and cytosolic NAD+ [3], and others. We measured Ca2+-induced mitochondria swelling, which represents a well-established and commonly accepted method of evaluation of MPTP opening in isolated mitochondria [4]. The measurements using this technique correlate with the [3H]DOG technique. Indeed, in the present work, no cytochrome c release or caspase activation were studied. Cytochrome c release from the mitochondria and activation of caspase cascade leading to apoptosis can be a result of MPTP opening. These important parameters need to be investigated in future experiments to study the mechanism of ischaemia/reperfusion injury and 8-Br-induced cardioprotection. We added this information to the Methods (Lines 203-207) and Discussion (Lines 511-513).
- It has been suggested in the study by Pereira et al (2020) in the PLOS ONE journal that a number of other factors in addition to HKII binding to the mitochondrial membrane protect against mitochondrial permeability transition pore opening. The authors should include this discussion point in the paper.
Response:
We thank the reviewer for this suggestion. This is an important article for our Discussion. Pereira et al. have demonstrated that in vitro dissociation of HKII alone does not fully replicate the effects of global ischaemia on mitochondrial function [5]. The auithors concluded that in vivo dissociation of HKII regulates mitochondrial function at the end of ischaemia indirectly, probably by interaction with mitochondrial fission proteins such as Drp1 and Drp2. They hypothesized that the changes in cristae structure and mitochondrial morphology occurred due to this interaction destabilize the integration between the outer and inner membrane, resulting in increased cytochrome c release and sensitizing MPTP to Ca2+ overload. We added this information to the Discussion (Lines 497-505).
- This is a study undertaken in rats and it is uncertain whether these results would translate into human studies. Some discussion about this fact would be pertinent in the manuscript.
Response:
The reviewer is correct. The results obtained in experiments on small rodents, including those on the pathological models, cannot be directly extrapolated to humans [6]. To advance the translation of our results to clinical practice, the protective effects of 8-Br need to be tested on a large animal model such as a pig, in which morphological and physiological characteristics of the cardiovascular system are closely matched to the humans [5]. Over the years, we have developed expertise in porcine models of cardiac surgery with cardio-pulmonary bypass and cardioplegic arrest [7,8]. This experience can be employed should the experiments on pathological in vivo and ex vivo models in rats confirm the potent cardioprotective effect of 8-Br. We added this information to the Discussion (Lines 519-527).
References
- Briston, T.; Roberts, M.; Lewis, S.; Powney, B.; Staddon, J.M.; Szabadkai, G.; Duchen, M.R. Mitochondrial permeability transition pore: sensitivity to opening and mechanistic dependence on substrate availability. Sci Rep 2017, 7, 1-13.
- Javadov, S.A.; Clarke, S.; Das, M.; Griffiths, E.J.; Lim, K.H.; Halestrap, A.P. Ischaemic preconditioning inhibits opening of mitochondrial permeability transition pores in the reperfused rat heart. J Physiol 2003, 549, 513-524.
- Di Lisa, F.; Menabò, R.; Canton, M.; Barile, M.; Bernardi, P. Opening of the mitochondrial permeability transition pore causes depletion of mitochondrial and cytosolic NAD+ and is a causative event in the death of myocytes in postischemic reperfusion of the heart. J Biol Chem 2001, 276, 2571-2575.
- Wong, R.; Steenbergen, C.; Murphy, E. Mitochondrial permeability transition pore and calcium handling. In Mitochondrial Bioenergetics, Springer: 2012; pp. 235-242.
- Pereira, G.C.; Lee, L.; Rawlings, N.; Ouwendijk, J.; Parker, J.E.; Andrienko, T.N.; Henley, J.M.; Halestrap, A.P. Hexokinase II dissociation alone cannot account for changes in heart mitochondrial function, morphology and sensitivity to permeability transition pore opening following ischemia. PloS one 2020, 15, e0234653.
- Ludman, A.J.; Yellon, D.M.; Hausenloy, D.J. Cardiac preconditioning for ischaemia: lost in translation. Dis Model Mech 2010, 3, 35-38.
- Alvino, V.V.; Fernández‐Jiménez, R.; Rodriguez‐Arabaolaza, I.; Slater, S.; Mangialardi, G.; Avolio, E.; Spencer, H.; Culliford, L.; Hassan, S.; Sueiro Ballesteros, L. Transplantation of allogeneic pericytes improves myocardial vascularization and reduces interstitial fibrosis in a swine model of reperfused acute myocardial infarction. J Am Heart Assoc 2018, 7, e006727.
- Gadeberg, H.C.; Bond, R.C.; Kong, C.H.; Chanoit, G.P.; Ascione, R.; Cannell, M.B.; James, A.F. Heterogeneity of T-tubules in pig hearts. PLoS One 2016, 11, e0156862.